# Ascorbic Acid and Ozone: Novel Perspectives to Explain an Elusive Relationship

**DOI:** 10.3390/plants8050122

**Published:** 2019-05-09

**Authors:** Erika Bellini, Mario C. De Tullio

**Affiliations:** 1Department of Biology, University of Rome ‘Tor Vergata’, 00133 Rome, Italy; erikabellini1990@gmail.com; 2Department of Biology, University of Pisa, 56126 Pisa, Italy; 3Department of Earth and Environmental Sciences, University of Bari, 70125 Bari, Italy

**Keywords:** ascorbate redox balance, apoplast, hormesis, epigenetics

## Abstract

A huge amount of studies highlighted the importance of high ascorbic acid (AA) content in ozone tolerance, yet the relationship between them appears more complex than a simple direct correlation. Sometimes the connection is clear, for example, two *Arabidopsis* mutants defective in the main AA biosynthetic pathway (*vtc* mutants) were identified by means of their ozone sensitivity. However, some low-AA containing mutants are relatively tolerant, suggesting that AA location/availability could be more relevant than total content. A clear distinction should also be made between ozone tolerance obtained when AA content is increased by experimental supplementation (exogenous AA), and the physiological role of plant-synthesized AA (endogenous AA), whose amount is apparently subjected to tight regulation. Recent findings about the role of AA in signal transduction and epigenetic regulation of gene expression open new routes to further research.

## 1. Introduction

The origin of research on plant ozone (O_3_) sensitivity dates back to the late 1950s, when a leaf spot disease known as weather fleck was identified as caused by ozone [1]. Early experiments evidenced a protective role of ascorbic acid (AA) against O_3_ injury in Pinto bean plants [2]. Since then, several papers supported the view of a direct correlation between AA content and ozone tolerance [3,4]. Over the years and decades, this notion has led to the conclusion that ozone tolerance in a given plant species or cultivar could be somewhat anticipated on the basis of its AA content [5]. We present here some considerations on the relationship between AA and plant ozone tolerance, without the pretension of offering a complete and exhaustive review of the wide literature addressing the mechanisms of plant ozone tolerance (a limited choice of suggestions for further reading is offered in References [6,7,8,9,10,11,12,13,14,15]). Many accurate studies clearly showed that higher AA content is not the only possible mechanism involved in ozone tolerance [4,16], and that the AA/ozone relationship is more complex than one might expect on the basis of a simple interaction between reactive oxygen species (ROS) and antioxidants facing each other. Some relevant points will be critically discussed in the next sections, starting from a few basic questions: Is AA content always directly related to ozone tolerance? Is AA cellular location involved in ozone tolerance? What is the relationship between exogenous (experimentally supplied) and endogenous (synthesized by the plant) AA in O_3_ tolerance? What do we know about the mechanisms of AA-mediated tolerance? Recent data about AA involvement in the epigenetic control of gene expression opened the way to so far unexplored possibilities of explaining contrasting results obtained in decades of experimental work.

## 2. Is AA Content Always Related to Ozone Tolerance?

In some cases, a direct relationship between low AA content and increased sensitivity to O_3_ was observed. An outstanding example of this correlation is the identification of the AA-deficient *Arabidopsis thaliana* mutant *vtc1*, containing only 25–30% of total AA content (compared to the wild type), and identified on the basis of its ozone sensitivity [17]. Notably, this mutant had been initially described as *soz1* (*sensitive to ozone*) [18], and then renamed after further investigation leading to the characterization of the mutation in the key AA biosynthetic gene encoding GDP-mannose pyrophosphorylase [19]. However, the AA/O_3_ connection is not always proved. In an early attempt aimed at identifying the mechanisms of ozone tolerance in the tobacco cultivar Bel-B in comparison to the hypersensitive cv. Bel-W3, Menser could not find any obvious difference in AA content between the two cultivars, in spite of marked differences in the visible leaf injury caused by ozone [20,21]. AA content increased roughly to the same extent in both tobacco cultivars when plants were subjected to a 12 h light pre-treatment before fumigation, but leaf damage, although much reduced in comparison to control plants kept in the dark, was still higher in the sensitive Bel-W3 cv. than in the tolerant Bel-B exposed to the same light conditions [20]. AA supplementation to detached leaves effectively increased AA content, but still the extent of leaf injury in the two cultivars was not proportional to AA content [20]. Young leaves generally showed higher AA content, and less symptoms (Figure 1), but this relationship was less clear in the sensitive cv. Bel-W3 [20]. Data reported in the Menser paper [20] have been confirmed by a large number of similar observations [3,4,22,23], in *Plantago major* cultivars [16,24], in snap bean ecotypes [25,26], common bean [27], soybean cultivars [22,28], broad bean [29], *Sedum album* [30], *Nicotiana* sp. [31], and in radish [4].

Summarizing, the main observations in support of a direct relationship between AA content and ozone tolerance are:Exogenous AA induces protection against ozone injury.Light pre-treatment increases AA content and improves ozone tolerance.Young tissues generally have more AA and are less affected by ozone.

On the other hand, differences in AA content do not always correlate with different ozone sensitivity, and in some cases lower AA occurs in O_3_-tolerant plants. This result was especially clear when Conklin et al. identified additional AA-deficient *Arabidopsis vtc* mutants [32] in the follow-up of their work on the ozone-sensitive, low-AA *vtc1* mutant [17,18]. Their attempts to use ozone sensitivity as a screening tool to identify AA-deficient mutants were almost unsuccessful: In their screening of a population of 100,000 mutagenized plants, only two mutants (including the previously characterized *vtc1*) were, at the same time, ozone-sensitive and AA-deficient, accounting for less than 5% of the pool of ozone-sensitive plants observed in the screening [32]. This means, quite surprisingly, that over 95% of the ozone-sensitive mutants screened were not affected in their AA content. Moreover, when additional low-AA containing *Arabidopsis* mutants were isolated using a different screening method and their ozone sensitivity was tested, it turned out that the two allelic variants of the *vtc1* mutation were both ozone sensitive, but one of the *vtc2* alleles (*vtc2-2*) was unaffected by ozone treatment, whereas *vtc3* and *vtc4* were only slightly affected (Figure 2). All in all, neither high AA content in itself is a guarantee of ozone tolerance, nor is low AA synonymous with sensitivity. The mechanism underlying the apparent AA-related protection from ozone injury should be found elsewhere.

## 3. What Do We Talk About, When We Talk About “Total Ascorbic Acid”?

The general debate on antioxidants, and more specifically on the role of AA in any physiological phenomenon (including ozone tolerance), is complicated by a certain degree of uncertainty regarding the actual identity of the molecules we are taking into consideration when talking about (or measuring) AA. It should be considered that AA is the non-dissociated form of the molecule. In the cellular environment, the anionic form ascorbate is definitely more represented, depending on the pH of the different compartments and organelles. One-electron oxidation of AA results in the production of the unstable radical AFR (Ascorbate Free Radical, or monodehydroascorbate). AFR dismutation gives both AA and the double oxidized form dehydroascorbic acid (DHA) [33]. The latter can be reduced back to AA both enzymatically (via the glutathione-dependent DHA reductase), or non-enzymatically. Although AA and DHA are very different in reactivity, the sum of the two is often referred to as “total AA”, because it is still possible to recover AA from DHA, whereas the next step of DHA catabolism, producing diketogulonic acid, is irreversible. However, to have full DHA reconversion to AA one should assume an extremely efficient DHA reductase activity, which is apparently not the case [34,35]. Not even plants overexpressing DHA reductase achieve the full conversion, although in such plants AA content is actually increased and results in higher ozone tolerance [23]. Therefore, measuring total AA appears of limited use, and the two species should be measured separately. In early studies, proper quantitation of AA and DHA was difficult to achieve due to technical limitations of the methods used. In addition, AA oxidation to DHA occurs spontaneously, and can even be accelerated under inadequate experimental conditions. In many studies reporting quantitative data on AA and DHA content under different stress conditions, statistical analyses showed high variability in biological replicates. In spite of such limitations, the AA redox status (expressed as the AA/DHA ratio, or sometimes as AA/AA + DHA) is potentially useful as a tool to investigate the mechanisms of plant responses to stress conditions.

## 4. AA and ROS Scavenging in Relation to Ozone Tolerance/Sensitivity

Being a non-polar molecule, ozone penetrates plant leaves through the stomata, and dissolves into the apoplastic fluid [36,37]. In this microenvironment it is intensively reactive, and spontaneously decomposes into reactive oxygen species (ROS) and/or reacts with a number of cell wall/apoplastic fluid/plasma membrane components to produce ROS, including superoxide radicals (O_2_^•−^), hydrogen peroxide (H_2_O_2_), and the highly reactive hydroxyl radical (OH^•−^) [38,39]. ROS are physiologically generated during cell metabolism and are involved in many biological processes including long-distance signaling, abiotic and biotic stress, cell death, and symbiosis [40,41,42,43,44,45,46,47].

The identification in plants of a large number of antioxidant molecules (AA, tocopherols, glutathione) and enzymes (AA peroxidase, superoxide dismutases, catalases, and so on) potentially able to reduce ROS to less harmful chemical species, supported the view that the antioxidant defense system plays a fundamental role in keeping ROS level under control [48]. AA always received special attention as the main candidate in the ROS detoxification mechanism. Direct (non-catalyzed) reaction with ROS causes AA oxidation to DHA [33]. Alternatively, DHA is produced in the reaction catalyzed by the hydrogen peroxide-reducing enzyme AA peroxidase, via AFR dismutation [49]. AA peroxidase isoenzymes, which are present in most subcellular compartments within the plant cell and are responsible for the conversion of hydrogen peroxide into water [49], are possibly involved in ozone tolerance. In fact, some studies evidenced high cytosolic AA peroxidase mRNA levels upon ozone exposure [50,51,52,53]. Furthermore, transgenic tobacco plants expressing an antisense cytosolic AA peroxidase showed high susceptibility to ozone injury [54]. However, overexpression of both cytosolic and chloroplastic AA peroxidase in an ozone-sensitive tobacco line neither conferred protection against ozone injury, nor increased sensitivity [55].

Much research focused on AA content in the apoplast, which is considered the first line of defense against O_3_-derived ROS. Although most AA is localized in the cytoplasm, up to 10% of the AA content of the whole leaf is exported and localized in the apoplast, where it can be found at millimolar concentrations [48]. Early studies suggest that the pool of apoplastic AA is an important component of the defense against ozone injury [30,56,57]. Ranieri and colleagues [58] observed an increase in both the apoplastic and intracellular AA level in both young (asymptomatic) and mature (symptomatic) leaves of pumpkin plants exposed to 150 ppb O_3_ (5 h d^−1^; 5 days), thus supporting the hypothesis of an O_3_-induced stimulation of AA synthesis followed by active export to the apoplast. Moreover, mathematical modelling suggests that, in general, the concentration of apoplastic AA should be sufficient to directly detoxify the majority of ozone absorbed into the leaf [59]. In a study on the spinach plant, young leaves containing highest apoplastic AA concentrations did not exhibit the necrotic leaf damage typical of ozone exposure. In contrast, the same study highlighted the relevance of AA redox status in long-time fumigation with ozone. Although initially the concentration of reduced AA did not change in the apoplast, later on the oxidized form (DHA) increased and the reduced form remained at a very low constant level [56]. Another independent study on *Arabidopsis thaliana* reported that AA in leaf apoplast extracts was almost entirely oxidized after ozone treatment, suggesting that O_3_ scavenging by direct reactions with reduced AA was very limited. Conversely, the same study shows that O_3_ stimulated transcription of several phenylpropanoid pathway genes, increasing the apoplastic concentration of sinapoyl malate [60]. High levels of extracellular AA in the oxidized form were also detected in both ozone-tolerant and ozone-sensitive soybean plants, suggesting that, in this case, extracellular antioxidant molecules different from AA are responsible for ROS detoxification and O_3_ tolerance [61].

In some cases, the AA/DHA ratio in the apoplast closely matches plant ozone tolerance or sensitivity. This was observed in transgenic plants showing altered AA redox status. Tobacco plants overexpressing DHA reductase not only had higher AA content but also lower DHA, thus switching the AA/DHA ratio from 3.4 to 9.3 [23]. Knocking down the DHA reductase gene lowered the AA/DHA ratio to 1.8. Overexpressors were more tolerant to ozone than wild-type plants, whereas knockout plants showed increased sensitivity [23]. Conversely, overexpression of the gene encoding the apoplastic enzyme AA oxidase, by lowering AA content and increasing apoplastic DHA, increased ozone sensitivity in parallel with lowering the AA/DHA ratio [31]. However, D’Haese et al. [62] observed a higher AA redox status in an ozone-sensitive clone of *Trifolium repens* when compared to a tolerant one. These apparently surprising data suggests that apoplastic AA, rather than quantitatively acting in ROS removal, could be related to signaling [63], as will be discussed in the next section of this article.

## 5. The Role of Ascorbate in the Signal Transduction Pathway

AA is ubiquitous in eukaryotes. Lack of dietary AA in the few organisms unable to synthesize it leads to the deadly syndrome known as scurvy, caused by the inactivation of some key AA- and oxoglutarate-dependent dioxygenases (2-ODDs) involved in the synthesis of collagen and other relevant molecules [64,65]. Notably, the loss of biosynthetic capability in primates and a few other mammals derives from the accumulation of mutations in the gene encoding l-gulono-lactone oxidase, the protein catalyzing the final step of the biosynthetic pathway [65]. Therefore, from an evolutionary point of view, biosynthetic capability is the original (plesiomorphic) state, and its loss is the derived (apomorphic) condition. Plants can synthesize AA and are the main source of this compound for non-synthesizing animals. Some AA-dependent enzymes related to 2-ODDs, including peptidyl-prolyl-hydroxylase, are present in both animals and plants [65], whereas some others are plant-specific. Among them, violaxanthin de-epoxidase (involved in the xanthophyll cycle) [66], myrosinase (hydrolyzing glucosinolates) [67], ACC oxidase (catalyzing the last step in ethylene biosynthesis) [68], and a few hydroxylases involved in gibberellin metabolism [69].

In the last 3–4 decades, much attention has been given to the antioxidant role of AA, i.e., the complex network of reactions involving AA and ROS. For years we have been exposed to many simplistic pieces of research, based on the assumption of a contraposition between ROS and antioxidants. Later on, novel observations substantiated the hypothesis that ROS and antioxidants are involved in signaling and the regulation of gene expression [63,70]. It is now clear that ROS act as environmental signals, guiding plant responses to unfavorable conditions. Therefore, the role of antioxidants, and AA among them, cannot just be the removal of toxic ROS, as often claimed in the past. 

It is noteworthy that plants respond to several stress conditions with a biphasic response, i.e., low levels of the stressor induce optimal plant growth, whereas higher levels of the same stressor are detrimental for the plant. This phenomenon is known as hormesis [71,72]. A form of hormesis known as pre-conditioning occurs when organisms previously acclimated to low-stress conditions are then exposed to high stress, to which they respond much better than unacclimated ones [73]. The relationship between ROS and antioxidants in establishing the hormetic response in animals under hypoxic conditions and re-oxygenation has been discussed by Oliveira et al. [74]. In this interesting paper the authors identify ROS and other molecules as the primary hormetic signal and advance some hypotheses on the molecular mechanism beyond the response.

Hormesis is likely to have an epigenetic basis [75], and recent experimental evidence confirms the involvement of epigenetic mechanisms in the regulation of hormetic responses [76], with fascinating consequences on our understanding of the phylogenesis of all living organisms [77]. Epigenetics is a key mechanism in building the “memory” of plants, a way for storing information that can be used later on by the same individual that perceived the stress, or by the next generations [78]. Interestingly, plants react to ozone treatment with a biphasic, hormetic dose-response behavior [79], and retain memory of ozone stress [80,81,82]. The regulation of gene expression via epigenetic mechanisms is likely to be the endpoint of several different signaling pathways. Indeed, AA can be involved in signaling in at least three different ways (Figure 3), all converging to epigenetic mechanisms. The first one is related to redox regulation [83], i.e., the balance between ROS and antioxidants. The second one is possibly mediated by plant hormones [84], including ethylene and gibberellins, whose synthesis requires AA. The third one directly involves the requirement of AA in the demethylation of nucleotides and histones. Demethylation of methylcytosine requires the family of enzymes known as Ten-Eleven-Translocation (TET) [85]. TET enzymes have been initially identified in mammal cells, but it is not yet clear whether they are actually present in plants. TETs can catalyze the hydroxylation of methylcytosine with the typical mechanism of 2-ODDs (Figure 4). Oxoglutarate is also required for the catalytic mechanism. In this reaction, AA is not just a generic reductant required to keep iron in the Fe^2+^ form, since other potential reductants are not effective in regulating the functioning of the enzyme. Further steps catalyzed by the same TET proteins can lead to cytosine demethylation and consequent reprogramming of gene expression [85].

The presence of an appreciable level of hydroxymethylcytosine in plant cells has been questioned [86], but further studies detected it in significant amounts in rice cultivars [87]. Remarkably, early studies in mammal cells failed to detect hydroxymethylcytosine in several cell types, although it is now clear that the hydroxylation of methylcytosine is widespread in animals [85]. Methylation of lysine and arginine in histone proteins is another mechanism of epigenetic control [85]. The mammalian histone demethylase JHDM1 operates by removing methyl groups from specific lysine residues of histone proteins, following the 2-ODD mechanism [88]. Similar histone demethylases characterized by the Jumonji C (JMJC) signature are also present in plants [89], and their involvement in plant responses to stress conditions is confirmed by recent reports [90,91].

## 6. Conclusions

Increased tropospheric ozone is likely to be one of the key factors that will shape the future vegetation of our planet [10,11,92]. The identification of the mechanisms of ozone tolerance could help in improving crop tolerance and avoid losses in productivity.

Although AA is often considered associated with ozone tolerance, critical reappraisal of available experimental data provides more doubts than certainties. The degree of ozone tolerance is not obviously related to endogenous AA content. On the other hand, quite a lot of studies reported that exogenous AA supply improves plant performance under ozone stress. It should be considered that the basal AA content in AA-synthesizing organisms is apparently tightly regulated at both the transcriptional, translational, and post-translational levels [93]. This is in contrast to the widespread concept that unlimited AA synthesis would be desirable, in order to counteract the toxic effects of ROS. The apparent contradiction between the undisputed importance of AA in cell metabolism, and the partial or total limitations in its synthesis can only be explained considering AA as a signaling molecule, rather than a bulk ROS scavenger. Signals convey information, and the effectiveness of a signal (i.e., its capability to transfer information) is lost whenever the signal itself goes far above the saturation level. When AA content is transiently increased by external supply (exogenous AA), plants appear more tolerant to ozone injury. However, in the long run such high AA content is likely to make AA signaling ineffective. Frei and co-workers [94] showed that the disruption of the AA biosynthetic gene gdp-mannose 3′-5′ epimerase in a rice insertion mutant caused, in parallel with a 20–30% decrease in AA content, a re-organization of the whole antioxidant system. We hypothesize that AA content regulates the baseline of plant stress responses, possibly by directly affecting gene expression via epigenetic mechanisms. Interestingly, AA-deficient rice did not show increased sensitivity to realistic ozone concentrations, whereas mutant plants were more affected than controls at high ozone values [94]. It remains to be elucidated whether AA is involved in plant recovery after ozone stress.

The working model described in Figure 5 provides a tentative explanation to the variety of experimental data discussed in the previous sections. In order to cope with unfavorable conditions, plants first have to sense the environment, and in particular those information-conveying molecules that are formed as a consequence of usual metabolic activities: Both ROS and AA are obvious candidates in this role. If ROS-scavenging capability is artificially altered, either by increasing AA content or by overexpressing AA peroxidase activity, or both, ROS-dependent visible signs of ozone injury are decreased, but the signaling mechanism is possibly also affected.

In conclusion, the AA/ozone connection, if explored avoiding tempting oversimplifications [95], appears a promising field for future research aimed at understanding the general mechanisms of plant stress responses.

## Figures and Tables

**Figure 1 plants-08-00122-f001:**
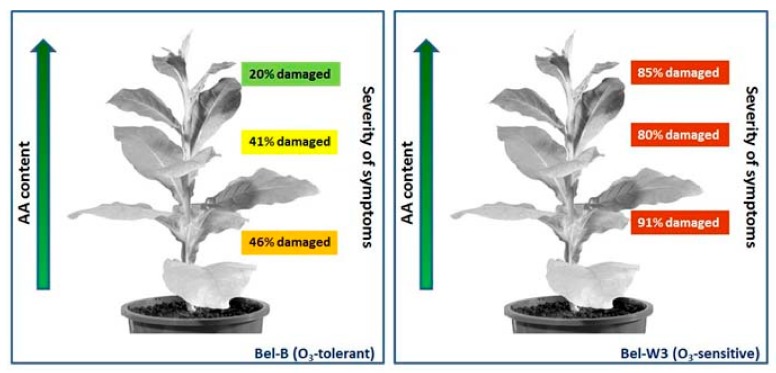
Relationship between ascorbic acid (AA) content and leaf injury in two tobacco cv. differing in ozone sensitivity. Scheme based on data from Menser [20].

**Figure 2 plants-08-00122-f002:**
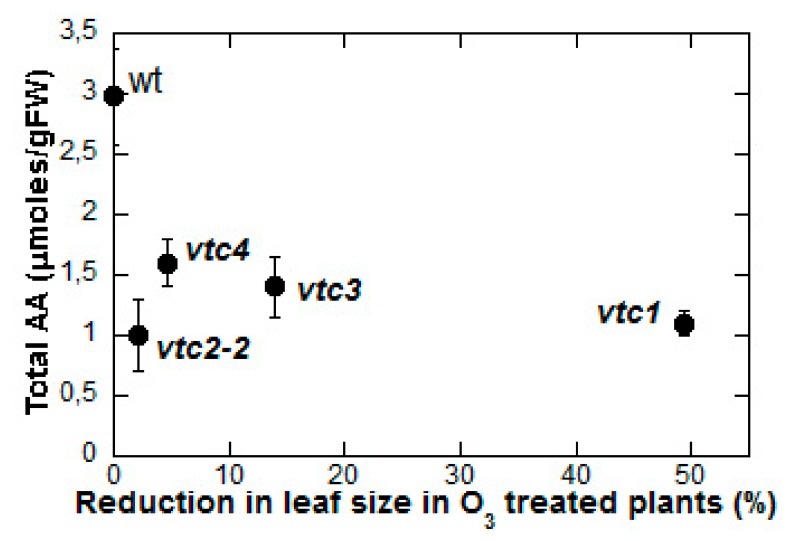
Ozone sensitivity in *Arabidopsis* mutants with low AA content. Graph based on data reported by Conklin et al. [32].

**Figure 3 plants-08-00122-f003:**
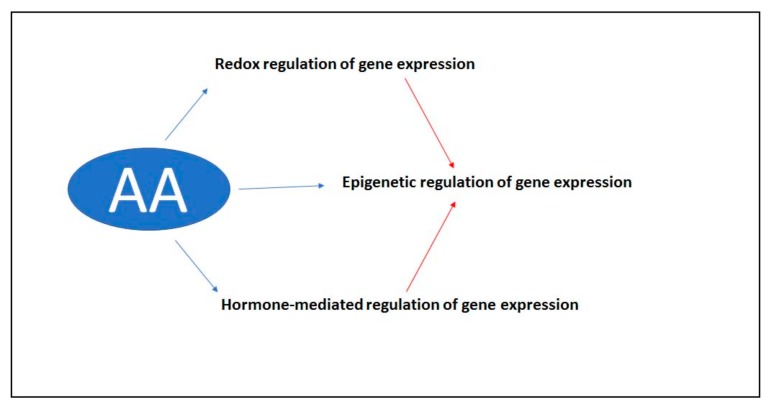
The possible action of AA in plant signal transduction and epigenetic responses.

**Figure 4 plants-08-00122-f004:**
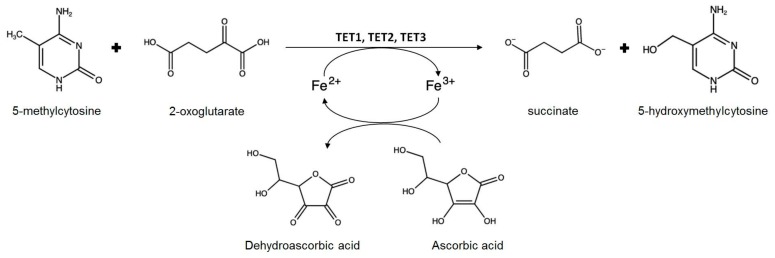
Role of AA in the hydroxylation of methylcytosine.

**Figure 5 plants-08-00122-f005:**
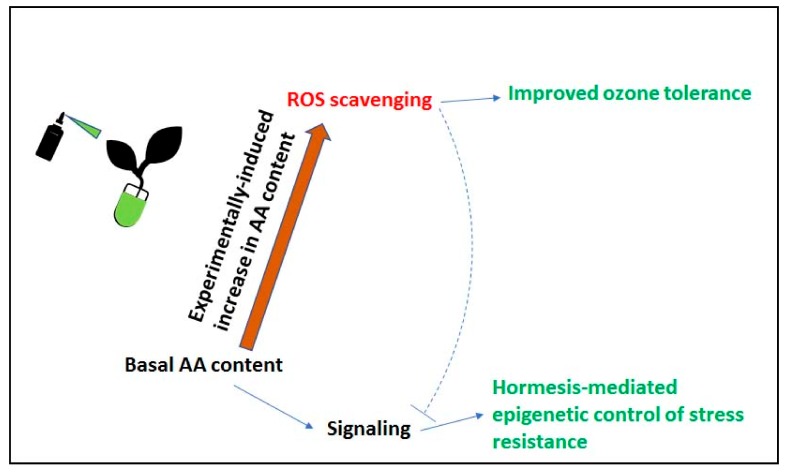
Scheme illustrating the two possible roles of AA in ozone tolerance. Treatments increasing AA content improve plant ROS-scavenging capability, reducing ozone-induced symptoms (upper part of the scheme). In the absence of treatments (lower part of the scheme), endogenous AA is used for AA-dependent signaling, which results in epigenetic modifications possibly involved in general stress responses.

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
