# Peer review of "Ascorbic Acid and Ozone: Novel Perspectives to Explain an Elusive Relationship"

_plants, 2019, doi:10.3390/plants8050122_

Round 1
Reviewer 1 Report
This manuscript aims to review the relationship between Ascorbic Acid content and the O3 resistance expressed by the plants.
Even though there is quite a lot of information about the ameliorative role of AA against different stressors, this paper adds the novel perspectives on this kind of research. I believe that this Manuscript can be accepted by MDPI-plants after making the suggested corrections.
Specific comments:
All through the paper, the term O3 resistance should be used instead of the O3 tolerance.
Abstract: A clarification of endogenous or exogenous AA content should be given.
Introduction: The same as above. Also an additional question could be: What is the relationship between endogenous or exogenous AA to O3 resistance?
References: The old bibliography should be avoided, eg years 1959, 1960, 1964, 1988, 1989.
It would be interesting if studies and discussion about the role of AA to O3 stress recovery could be referred in the paper.
Author Response
All through the paper, the term O3 resistance should be used instead of the O3 tolerance.
According to the suggestion of the Academic Editor, we used "tolerance" and "tolerant" throughout the manuscript
Abstract: A clarification of endogenous or exogenous AA content should be given.
The following sentence was added
A clear distinction should also be made between the effects of AA in ozone tolerance when AA content is increased by experimental supplementation (exogenous AA), and the physiological role of plant-synthesized AA (endogenous AA), whose amount is apparently subjected to tight regulation.
Introduction: The same as above. Also an additional question could be: What is the relationship between endogenous or exogenous AA to O3 resistance?
The question was added. The exogenous/endogenous relationship is discussed in the final section.
References: The old bibliography should be avoided, eg years 1959, 1960, 1964, 1988, 1989.
Older references were removed, unless they are historically relevant.
It would be interesting if studies and discussion about the role of AA to O3stress recovery could be referred in the paper.
This is an excellent comment. To our knowledge, this point has not been thoroughly investigated with specific experiments. Available evidence seems to support a role for AA in stress recovery. A sentence on this was added in the discussion.
Reviewer 2 Report
There are several errors, including incorrect use of verbs. The introduction is pretty bad. I find it too insufficient While there is some discussion of the role of ascorbic acid, there is no introduction at all of the apoplast, epigenetics and hormesis which are mentioned in the keywords too. The current introduction is far from being a proper introduction of a review paper. It feels that authors drafted this section in a highly rush manner within 10 minutes without critical thinking. It even over-simplifies the current state-of-art of the research on and role of ascorbic acid. There are also some misleading statements such as in lines 27-31. There are huge efforts dealing with the role of ascorbic acid both in tree and crop research and far beyond “supplementing plants with AA before ozone fumigation”. Not only comparisons of the role of ascorbic acid have been done at species level, but they have been also done at genotype level. The literature should be thoroughly surveyed and reviewed. For example, see works by the groups of Dr. Burkey (USDA), Prof. Agrawal (India) and Dr. Frei (Germany) for crops and by the groups of Prof. Oksanen (Finland), Dr. Nali (Italy) and Prof. Izuta (Japan). Furthermore, the first two questions given in lines 32-33 are presented as being unanswered, however, these are well known and documented not only in research papers but also in a series of review papers, including meta-analyses. Several of them were published in the last three years. A check of the following sections and reference list reveals that none of them was discussed or even cited. A tremendous body of research regarding the role of ascorbic acid in ozone stress is totally ignored, hence, the paper being non-representative of the current state-of-art. A review paper should be critical. What is the critical point of the paper and what is new? I regret to say that section 2 is worse than section 1. This section is far below par publication standards of review articles. It gives a surprisingly narrow view of a limited number of papers (among dozens of hundreds) which can be misleading. The title of the section does not accurately reflect its content. This section is mainly conserved with supplementation of ascorbic acid. This section does not represent or consider endogenous production of ascorbic acid under ozone stress and I doubt this is the proper way to do as it would then be a matter of species or plant-specific supplied concentration, dose or dosage, being affected by a series of other methodological conditions. In some cases, discussion is incomplete, thus, not providing the full image. For example, see lines 46-49. What was the light source? How much was the light? Was this a type of preconditioning? Was the light natural sun light? See also the next lines (49-51). How do detached leaves support the conclusion at plant level (role of ascorbic acid in plant resistance to ozone). In line 64, light pre-treatment-induced content of ascorbic acid is assumed to improve ozone resistance, and this is given as a summary of the section based on only one reference. Discussion in lines 66-80 of the same section is very poor/incomplete (see dozens of papers from several groups around the globe). Figure 2 is useless. Section 3 is just an interesting one-paragraph discussion but cannot be a stand-alone section. Some parts of section 4 are too introductory and they should rather be in the Introduction. Section 4 is incomplete as well. Please properly survey relevant literature. In section 5, the argument in lines 188-189 is vague, false and misleading. Please refer to a series of review papers published in between 2016 and 2018 regarding the role of ROS in plants and animals. The sentence in lines 192-194 is not supported by any reference, and, given the too generalized assumption a series of references should be cited to account for different stressors, plant species, and biological traits. Same applies to the sentence in lines 194-195. The statement in lines 195-196 is incorrect. This phenomenon is known as pre-conditioning, not as hormesis as mentioned. While pre-conditioning is a manifestation of hormesis, it is not hormesis itself. Please correct it and clearly explain. Regarding the conclusions, the reference cited in line 229 cannot support this argument. Stronger evidence is needed. In line 253, authors present Fig 5 as it is their self-introduced working model, however, these are already discussed in the literature. Please consider revising the phrasing. Regarding Fig 5, the caption should be enriched. This figure is unclear to me. It conveys the message that ascorbic acid induces signaling which results to hormesis –epigenetic control of stress resistance. While the hormetic function of ROS in plants and animals is extensively documented and discussed, I do not think that one can claim hormetic function of ascorbic acid, unless authors can provide evidence that ascorbic acid can be produced ENDOgenously at so high levels that can CAUSE adverse/toxic responses. Overall, I am supportive of the idea, however, the paper needs much of work –after a serious literature survey- before publication. The language needs improvement.
Author Response
The introduction is pretty bad. I find it too insufficient While there is some discussion of the role of ascorbic acid, there is no introduction at all of the apoplast, epigenetics and hormesis which are mentioned in the keywords too. The current introduction is far from being a proper introduction of a review paper. It feels that authors drafted this section in a highly rush manner within 10 minutes without critical thinking.
Before the Internet era, all review articles were necessarily exhaustive surveys, reporting (if not discussing) any publication related to a specific topic. Nowadays, a second form of review article (often referred to as “minireview”) offers less references, and possibly more ideas. To avoid misunderstandings, we added in the introduction the statement that we have no pretension of providing a fully referenced review. Anyway, we added some more references as suggestions for further reading, including papers by Agrawal, Frei, Oksanen and others. Stating that the introduction section was drafted in 10 minutes without critical thinking is pointless.
Furthermore, the first two questions given in lines 32-33 are presented as being unanswered, however, these are well known and documented not only in research papers but also in a series of review papers, including meta-analyses.
The questions are not "presented as being unanswered", as in the Reviewer's interpretation (prejudice). They are used to introduce some of the points addressed in the next sections.
I regret to say that section 2 is worse than section 1. This section is far below par publication standards of review articles. It gives a surprisingly narrow view of a limited number of papers (among dozens of hundreds) which can be misleading.
Thanks for the encouragement. Again, our goal is not listing hundreds of papers. We focused on a few representative articles.
Figure 2 is useless.
Figure 2 is useful, as it is the answer to the question in section 1 (is AA content always related to tolerance?).
Section 3 is just an interesting one-paragraph discussion but cannot be a stand-alone section.
Section 3 could make a separate “box” incorporated in the text. Unfortunately this is not compatible with the layout of Plants.
In section 5, the argument in lines 188-189 is vague, false and misleading. Please refer to a series of review papers published in between 2016 and 2018 regarding the role of ROS in plants and animals.
“For years we have been exposed to many simplistic pieces of research, based on the assumption of a contraposition between ROS and antioxidants.” What is false or misleading in this statement? Please see Scudellari, M. The science myths that will not die. Nature 2015, 528, 322-325.
The sentence in lines 192-194 is not supported by any reference, and, given the too generalized assumption a series of references should be cited to account for different stressors, plant species, and biological traits.
We accept this criticism. References have been added.
Same applies to the sentence in lines 194-195. The statement in lines 195-196 is incorrect. This phenomenon is known as pre-conditioning, not as hormesis as mentioned. While pre-conditioning is a manifestation of hormesis, it is not hormesis itself. Please correct it and clearly explain.
We agree on this. We briefly addressed the distinction between hormesis and pre-conditioning.
Regarding the conclusions, the reference cited in line 229 cannot support this argument. Stronger evidence is needed.
More references were added.
In line 253, authors present Fig 5 as it is their self-introduced working model, however, these are already discussed in the literature. Please consider revising the phrasing.
The sentence was modified.
Regarding Fig 5, the caption should be enriched.
Done.
This figure is unclear to me. It conveys the message that ascorbic acid induces signaling which results to hormesis –epigenetic control of stress resistance. While the hormetic function of ROS in plants and animals is extensively documented and discussed, I do not think that one can claim hormetic function of ascorbic acid, unless authors can provide evidence that ascorbic acid can be produced ENDOgenously at so high levels that can CAUSE adverse/toxic responses.
Although the initial idea of hormesis is strictly related to toxicology, the concept is now broader (see e.g. Calabrese and Mattson, How does hormesis impact biology, toxicology, and medicine? NPJ Aging Mech Dis. 2017; 3: 13. doi:10.1038/s41514-017-0013-z). Hypothesizing toxic effects of high AA doses is beyond our intentions. In our view, AA is not the hormetic signal in itself, rather it is involved in the molecular mechanism underlying the hormetic response. We rephrased the text to clarify this point.
Reviewer 3 Report
This manuscript by Bellini and De Tullio aims to review the role of ascorbic acid (AA) in ozone stress resistance. I felt that this manuscript is written for the specialized audience. I have few suggestions for the authors:
Addition of general information on ascorbic acid and how ozone stress impact plants in the introduction section will be helpful for general audience.
The connection of AA with the epigenetic control of stress resistance is not clear. Despite a well compiled literature, literatures are not well synthesized to establish the relationship between the role of AA and ozone stress resistance.
Figure content and quality of working model (Figure 5) could be improved. A more descriptive figure legend in this figure would be very helpful.
Few errors in references particularly italicizing the scientific names of the plant species.
Author Response
The connection of AA with the epigenetic control of stress resistance is not clear. Despite a well compiled literature, literatures are not well synthesized to establish the relationship between the role of AA and ozone stress resistance.
We tried to clarify this point in the Conclusion section
Figure content and quality of working model (Figure 5) could be improved. A more descriptive figure legend in this figure would be very helpful.
The figure was modified and the figure legend improved
Few errors in references particularly italicizing the scientific names of the plant species.
References were double checked
Reviewer 4 Report
The submitted review entitled “Ascorbic acid and ozone: novel perspectives to explain an elusive relationship” revealed the reported complex relationship between ascorbic acid (AA) and tolerance to ozone of plants: AA content is not always related to ozone-tolerance. Then the authors described the possible problems in measurement of AA: the sum of AA and dehydroascorbic acid (DHA) is often referred to as total AA. Finally the authors suggested their working model of roles of AA in ozone-resistance: AA can act as a simple ROS scavenger and also as a signaling molecule leading to epigenetic control of ozone-tolerance. This review is definitely suitable for the special issue “ozone tolerance mechanisms” of Plants, and is well-documented. For me, this version is acceptable for the publication in Plants, but I would suggest small modifications to improve readability as below:
In this review, two kinds of expressions, “ozone resistance” and “ozone tolerance”, were used. If the authors used the words “resistance” and “tolerance” to have the same meaning, consistent usage of one of the words may be preferable. I would suggest using “ozone tolerance” throughout the manuscript, since this review is submitted to the special issues of “ozone tolerance mechanisms”.
Figure 1: Please consider putting information to show that Bel-B is a tolerant cultivar and Bel-W3 is a sensitive cultivar in the figure or legend.
Author Response
In this review, two kinds of expressions, “ozone resistance” and “ozone tolerance”, were used. If the authors used the words “resistance” and “tolerance” to have the same meaning, consistent usage of one of the words may be preferable. I would suggest using “ozone tolerance” throughout the manuscript, since this review is submitted to the special issues of “ozone tolerance mechanisms”.
"Resistance/resistant" was replaced with "tolerance/tolerant" throughout the text
Figure 1: Please consider putting information to show that Bel-B is a tolerant cultivar and Bel-W3 is a sensitive cultivar in the figure or legend.
The figure was improved according to the Reviewer's suggestion.
Round 2
Reviewer 2 Report
Authors have revised the manuscript at some extend. At least it is more accurate in the revised version. I have no doubt the paper will attract high interest and from a wide audience, thus, accuracy is important to avoid misleading readers who will potentially be unfamiliar with some topics of the paper. I have seen Plants publish short updates on the latest progress made in a given area of research as Reviews, and I recommend the paper for publication after considering some points:
1) The issue was not the amount of references to be cited but what was introduced. Whatever mini-review (they existed before the internet era), longer critical review, or a systematic review, the reader should be “prepared” for what to expect. This could be addressed by adding just three too brief sentences. The authors have at least added some mentions now.
2) The questions have been improved at some extend.
3) Fig 2 was a typo for which I apologize. I was referring to Fig 3, thinking that this figure would be replaced by one very brief sentence. However, this is my personal opinion, it does not represent a scientific issue, and the journal is an open-access journal with no page limitations; thus, up to the authors/editors.
4) The issue of preconditioning-hormesis has been addressed. I agree on the change. However, a reference should be cited for guidance. You already cite some relevant references in your paper so you can just use some of them.
5) I agree on the revisions related to Fig 5. The caption has been improved significantly.
6) Authors corrected the text of Fig 5 by deleting hormesis, I should clarify a point that may help for conveying the correct message in the text. Hormesis is by definition a biphasic dose-response model (there are some papers by Calabrese and by Mattson and their colleagues on the definition). If you just say hormesis, it means both positive and negative effects. Therefore, “Hormesis-Epigenetic…” that was illustrated in the previous version of the figure would be less appropriate. “Hormesis-mediated epigenetic…” or something along this line would be appropriate and could be written. However, the hormesis-mediated process would begin before the epigenetics and thus in the figure would be illustrated for signaling too. The authors stated “In our view, AA is not the hormetic signal in itself, rather it is involved in the molecular mechanism underlying the hormetic response. We rephrased the text to clarify this point.” I agree with this view and they are very correct. While I know this would be the case, I should be objective. The antioxidants are obviously key players in hormesis, but it is necessary to be accurate with how something is stated. Regarding this, the work by Oliveira et al. (2018, Front Physiol, doi: 10.3389/fphys.2018.00945) would be interesting to be briefly discuss. They discuss in particular detail endogenous antioxidant enzymes, preparation for oxidative stress, and hormesis in animals. Their paper can strengthen the views of this paper as hormesis-based processes occur in all types of organisms.
è A note here just to avoid potential error in future publication (no need to do something related to this paper), the idea of hormesis was not “strictly related to toxicology” as you mention. The origins of hormesis are not even toxicology itself (there are dozens of papers on the historical foundations published in the late 1990s). The concept is broader, as you rightly mention, but for more than a decade (e.g. see Calabrese and Mattson 2010, DOI:10.1007/978-1-60761-495-1, and earlier works).
Minor editorial issues:
1) Line 280: change “know” to “known”.
2) Characterizations such as “slippery conclusion” line 28) are beyond the scope of a scientific paper and could be simply removed.
3) In Fig 1, the resolution of the text in the red boxes is pretty bad. It is too hard for me to read it and I guess I will not be the only one with this issue. Furthermore, it is confusing that the plant is exactly same in both panels but I understand it is difficult to do something with this.
Author Response
4) The issue of preconditioning-hormesis has been addressed. I agree on the change. However, a reference should be cited for guidance. You already cite some relevant references in your paper so you can just use some of them.
An appropriate paper by Calabrese (Pre-Conditioning is Hormesis, 2016 Pharmacol res.) was cited and added to the reference list.
6) Authors corrected the text of Fig 5 by deleting hormesis, I should clarify a point that may help for conveying the correct message in the text. Hormesis is by definition a biphasic dose-response model (there are some papers by Calabrese and by Mattson and their colleagues on the definition). If you just say hormesis, it means both positive and negative effects. Therefore, “Hormesis-Epigenetic…” that was illustrated in the previous version of the figure would be less appropriate. “Hormesis-mediated epigenetic…” or something along this line would be appropriate and could be written.
We changed the figure as suggested.
Regarding this, the work by Oliveira et al. (2018, Front Physiol, doi: 10.3389/fphys.2018.00945) would be interesting to be briefly discuss. They discuss in particular detail endogenous antioxidant enzymes, preparation for oxidative stress, and hormesis in animals. Their paper can strengthen the views of this paper as hormesis-based processes occur in all types of organisms.
The paper by Oliveira et al. was mentioned and briefly discussed.
Minor editorial issues:
1) Line 280: change “know” to “known”.
Thank you for spotting this typo.
2) Characterizations such as “slippery conclusion” line 28) are beyond the scope of a scientific paper and could be simply removed.
"Slippery" was removed
3) In Fig 1, the resolution of the text in the red boxes is pretty bad. It is too hard for me to read it and I guess I will not be the only one with this issue.
We agree. Both the background and the fonts in the boxes were changed to improve resolution
Reviewer 3 Report
The literature presented in this manuscript are still not convincing enough to suggest the role of ascorbic acid in epigenetic regulation during plant tolerance to ozone stress. The connection is more hypothetical than evidence-based. If this is authors’ hypothesis, more references are needed to support the hypothesis.
Author Response
The literature presented in this manuscript are still not convincing enough to suggest the role of ascorbic acid in epigenetic regulation during plant tolerance to ozone stress. The connection is more hypothetical than evidence-based. If this is authors’ hypothesis, more references are needed to support the hypothesis.
To our knowledge, a specific study addressing the effects of ascorbic acid on plant epigenetics in response to ozone is still lacking. We agree about the need of sound experimental data. Meanwhile we offer our hypothesis, in the hope of stimulating some more research in this area. We added references about the connection between hormesis and epigenetics.